# Current Advances in Papillary Craniopharyngioma: State-Of-The-Art Therapies and Overview of the Literature

**DOI:** 10.3390/brainsci13030515

**Published:** 2023-03-20

**Authors:** Gianpaolo Jannelli, Francesco Calvanese, Luca Paun, Gerald Raverot, Emmanuel Jouanneau

**Affiliations:** 1Skull Base and Pituitary Unit, Department of Neurosurgery B, Neurological Hospital Pierre Wertheimer, Bron, 69677 Lyon, France; gianpaolo.jannelli@chu-lyon.fr (G.J.);; 2Neurosurgical Unit, Faculty of Medicine, Geneva University Hospitals, University of Geneva, 1205 Geneva, Switzerland; 3Department of Neurosurgery, Helsinki University Central Hospital, Helsinki University, Meilahden tornisairaala, Haartmaninkatu 4 Rakennus 1, 00290 Helsinki, Finland; 4Department of Neurosurgery, GHU-Paris Psychiatrie et Neurosciences, Hôpital Sainte Anne, 1 Rue Cabanis, CEDEX 14, 75014 Paris, France; 5Department of Endocrinology, Neurological Hospital Pierre Wertheimer, University Hospital of Lyon, 69500 Lyon, France; 6Inserm U1052, CNRS UMR5286, Cancer Research Center of Lyon, University Claude Bernard Lyon 1, 69000 Lyon, France

**Keywords:** craniopharyngiomas, papillary type, BRAFV600E, CTNNB1, target therapy, adjuvant and neo-adjuvant treatment

## Abstract

Craniopharyngiomas are commonly classified as low-grade tumors, although they may harbor a malignant behavior due to their high rate of recurrence and long-term morbidity. Craniopharyngiomas are classically distinguished into two histological types (adamantinomatous and papillary), which have been recently considered by the WHO classification of CNS tumors as two independent entities, due to different epidemiological, radiological, histopathological, and genetic patterns. With regard to papillary craniopharyngioma, a BRAF V600 mutation is detected in 95% of cases. This genetic feature is opening new frontiers in the treatment of these tumors using an adjuvant or, in selected cases, a neo-adjuvant approach. In this article, we present an overview of the more recent literature, focusing on the specificities and the role of oncological treatment in the management of papillary craniopharyngiomas. Based on our research and experience, we strongly suggest a multimodal approach combining clinical, endocrinological, radiological, histological, and oncological findings in both preoperative workup and postoperative follow up to define a roadmap integrating every aspect of this challenging condition.

## 1. Introduction

Craniopharyngiomas (CPAs) are rare suprasellar tumors arising from the remnants of embryonic epithelial cells of the craniopharyngeal duct or from epithelial metaplasia at the level of the pituitary stalk. They represent 1.2–4.6% of all brain tumors with an incidence rate of 0.5–2.5 new cases per 1 million people [1].

Despite an overall survival rate of 89–94% at 5 years and 85–90% at 10 years follow-up, the high recurrence rate of CPAs often requires multimodal invasive treatments in order to obtain a good tumor control [1,2]. Indeed, the main problem in CPA management remains the low long-term quality of life, especially in the case of a tubero-infundibular and/or intraventricular location due to the close functional and anatomical relationship with the hypothalamus and neurovascular suprasellar structures [3,4,5,6,7,8,9].

CPAs are classically distinguished into two histological types, the adamantinomatous (ACP) and papillary (PCP) types. The first one accounts for 90% of all CPAs and presents a bimodal peak of incidence in childhood and in adulthood, whereas PCPs represent 10% of all cases and usually affect adult patients in the 4th–5th decade of life [1,10]. If in the past ACPs and PCPs were considered different variants of the same pathological entity, the 2021 WHO classification of CNS tumors stated that they represent two independent tumor types, due to different epidemiological, radiological, histopathological, and genetic patterns [11].

With regard to the genetic spectrum, the detection of Wnt/β-catenin alterations in ACPs, as well as the presence of BRAF V600E mutations in about 95% of PCPs, is opening new frontiers for targeted medical therapies [12]. If the results of molecular target vectors operating on the Wnt/b-catenin cascade are not yet encouraging, the current literature reports drastic tumor volume reduction associated with good clinical outcomes after oral targeted therapy with BRAF and MEK inhibitors in PCPs [1,13,14,15]. However, even if these findings seem to be very promising, many points, such as the secondary effects of this therapy, the duration of the treatment, and a possible role of a neo-adjuvant approach, remain unclear [16].

In this article, we perform an overview of the more recent progress made into understanding PCPs and the targeted therapies. We will particularly focus on the radiological and genetic spectra of PCPs in the two following paragraphs. Moreover, we will discuss the role of both adjuvant and neo-adjuvant approaches in PCP management with the aim of “tailoring” the more appropriate approach according to the different clinical and radiological settings.

## 2. Materials and Methods

The aim of this paper is to provide an extensive review of the medical therapies for treating PCPs. A systematic methodology has been applied only for this topic. New insights into the genetic and molecular aspects, as well as the radiological features of CPs are also offered in a purely descriptive manner to give a complete overview of the topic.

A search of the most relevant papers within the PUBMED, Google Scholar and MEDLINE databases was performed. All searches used the following keywords: ‘‘craniopharyngioma AND target therapy”, “craniopharyngioma AND medical treatment”, “craniopharyngioma AND papillary”, “papillary AND craniopharyngioma AND medical treatment”, “papillary AND craniopharyngioma AND BRAF inhibitors”, and “papillary AND craniopharyngioma AND BRAF/MEK inhibitors”.

Starting from the bibliographies of the articles found in our primary search, we performed a secondary search. Articles were reviewed by title and abstract for potential topic relevance, and if the titles or/and abstracts did not clearly indicate the degree of relevance, the articles were reviewed completely. The search was limited to human subjects and English language publications. Only full and relevant articles, as well as original communications were selected. Figure 1 shows the PRISMA flow diagram of the literature review.

The authors confirm that the data supporting the findings of this study are available within the article and from the corresponding author, upon reasonable request.

## 3. Results

Our primary search found 223 papers. Twelve papers were excluded because the full texts were not available. Following the exclusion of the duplicates, 25 articles were selected for their clinical and topic relevance.

Thirteen previously reported cases of PCPs treated with targeted therapy were found in the English language publications [16,17,18,19,20,21,22,23,24,25,26]. Among these, 12 reported adjuvant or rescue therapy for recurrent PCPs, and only one (i.e., a previous author publication) showed the result of a neo-adjuvant treatment for a newly diagnosed PCP. The preliminary data of a phase-2 clinical trial (NCT03224767) has been also reported and is discussed below in the “Adjuvant Treatment” section. Several other reports were excluded because they lacked clinical and/or radiological results. Only two specific surgical series dealing with PCPs were found [27,28]. The other studies were precedent literature reviews about a related topic [1,2,3,10,12,13,14,29,30,31,32].

## 4. Discussion

### 4.1. Radiological Features: Hypothalamic Invasion and Differential Diagnosis of ACPs/PCPs

CPAs present very heterogeneously upon routine imaging. Radiological features can vary from solid to cystic lesions, with or without calcifications, located in different anatomical regions [1,10].

Tumor topographical location and hypothalamic invasion at the level of the third ventricular floor (i.e., defined as the absence of a dissection plane between the tumor and the hypothalamus) represent the two most relevant features to evaluate in the radiological assessment [7]. This is due to particularly dramatic consequences on the quality of life for patients with postoperative hypothalamic injury [9]. Therefore, it is mandatory in the preoperative work-up to look for a potential hypothalamic involvement, as well as assess the riskless options to avoid any damage to this structure [30]. Different imaging-graded systems and radiological criteria have been proposed in the few last decades to predict pre-operative hypothalamic involvement and the risk of post-operative injuries in CPs. Puget et al. distinguished CPAs in three grades according to the degree of hypothalamic involvement: 0 = no hypothalamic invasion; 1 = compression of the hypothalamus that can still be identified; 2 = severe involvement or unidentifiable hypothalamus. The authors concluded that a gross total resection is possible in grade 0 and 1, while a subtotal tumor resection is safer when the tumor is grade 2. However, this classification in used for pediatric CPAs, and no similar system is available for adult cases [31]. Van Gompel et al. identified both hypothalamic changes on MRI T2WI and the irregular enhancement as pre-operative predictors of higher-grade hypothalamic involvement [20]. Hayashi et al. reported that the degree of the extent of the peritumoral hypothalamic edema on coronal FLAIR and T2WI MRI images could be an index of the pre- and postoperative functional outcome in adult patients [22,32,33]. Indeed, the authors found that patients with focal hypothalamic edema in MRI (group B) or extensive edema in the optic tract and internal capsule (group C) presented a worse outcome than those without edema (group A). Their conclusions correlate with Puget’s degree of hypothalamic involvement and confirms the results reported by Van Gompel et al. [34], Mortini et al. [22,35] and Higashi et al. [36]. An overview of the hypothalamic invasion criteria and the previously reported grading is synthetically shown in Table 1.

Particular tumor locations are also correlated with a higher risk of hypothalamus involvement. Pascual and Prieto topographically distinguished five categories of CPAs based on both the mamillary body angle and the relationship with the third ventricular floor: sellar/suprasellar, suprasellar/pseudo-intraventricular, suprasellar secondary intraventricular, tubero-infundibular, or not strictly intraventricular and ‘‘purely” intra-ventricular tumors [6,7,23]. The authors found that giant and complex tubero-infundibular or secondary intraventricular CPAs entail a higher risk of hypothalamic invasion and injury, regardless of the surgical approach [6,8,9]. This is explained by their intrinsic morphological characteristics. Indeed, tubero-infundibular tumors completely replace the third ventricular floor during their growth. Similarly, secondary intraventricular CPs extend from the sellar to the third ventricular, invading all structures in the vertical pituitary-hypothalamic axis.

Another crucial step is to radiologically differentiate PCPs from ACPs.

ACPs typically present as a mixed solid and macrocystic lesion located in the suprasellar cistern, with relatively large calcifications [29]. The solid component is heterogeneously enhanced after gadolinium enhancement. The cyst is usually large and hyperintense on T1, T2, and FLAIR weighted images due to the presence of proteinaceous liquid. Calcifications are detected in 90% of cases and better demonstrated on T2 imaging. Moreover, PCPs are more commonly solid or microcystic and are rarely calcified. Two recent studies investigated possible radiological features to distinguish ACPs from PCPs.

Sartoretti-Schefer et al. found that the spherical shape, hypointense cysts on T1-weighted images, and their predominantly solid appearance in PCPs are statistically differential manifestations from the lobulated shape, large hyperintense cysts on T1-weighted images of ACPs that also typically encase vessels [40]. With regard to the location, purely intrasellar lesions or isolated intraventricular lesions, as well as tubero-infundibular tumors seem to be generally associated with the papillary histological type. Yue et al. tried to predict the BRAF mutation status of CPAs using MRI features [41]. They found that BRAF-mutated CPAs tended to be suprasellar, spherical, predominantly solid, homogeneously enhancing, and showed a thickened pituitary stalk.

Despite the efforts to find some reliable criteria to be able to radiologically distinguish ACPs and PCPs, the radiological diagnosis remains challenging, and a tissue sample remains the only realistic method to obtain a definitive diagnosis.

### 4.2. Genetic Findings and Their Implication in Targeted Therapy

Several studies have found that ACPs and CPAs present two different clonal driver mutations with different implications, in terms of targeted therapy [12,29,42].

With regard to ACPs, a somatic mutation in the CTNNB1 gene is detected in a range of 70–96% of cases in different studies [1,21,25,43]. This range is probably explained by the different analytical method rather than the frequency of the mutation. In their original study, Brastrianos et al. found that the most frequent mutation is located at the level of the exon 3 degradation-targeting motif (51/53 tumors using genomic sequencing) [43].This mutation entails a higher stability and resistance to degradation of the β-catenin protein. Thus, the result is a hyperactivation of the Wnt/β-catenin pathway with a pro-tumorigenic senescence-associated secretory phenotype only in specific “cell clusters” and single cells through the tumor or at base of the finger-like protrusion [1,29]. Martinez-Barbera showed that the CTNNB1 mutation occurs only in SOX2+ pituitary tumoral stem cells which represent the “true” tumoral cells and lead to the tumor development in a paracrine manner. Single SOX2+ tumor cells or clusters of SOX2+ tumor cells are largely distributed in the tumor and could promote the proliferation and the invasion of the surrounding “non-mutated” epithelial cells by the overexpression of growth factors (vascular endothelial growth factor (VEGF), epidermal growth factor (EGF), fibroblast growth factor (FGF), and transforming growth factor beta (TGF-beta)), and cytokines (such as sonic hedgehog or SHH) [42,44]. The deregulation of the immune system could also play an important role in CP pathogenesis. Several studies have reported an increased expression of immunosuppressive factors (such as IL-10, galectin-1 and indoleamine-pyrrole 2,3dioxygenase) and pro-inflammatory cytokines (IL-1 and IL-6) in the tumoral and peritumoral microenvironment [1,45,46]. Coy et al. mapped and quantified the expression of the programmed cell death protein 1/programmed death-ligand 1 (PD-1/PD-L1) in 21 ACPs and 18 PCPs [47]. They found that PD-L1 is predominantly localized to the cyst lining in ACPs, and to the basally oriented tumor cells circumferentially surrounding the fibrovascular stroma in PCPs. Given their microstructural location and immunomodulatory function, PD1 and PDL-1 proteins could play a relevant role in tumor proliferation and infiltration. Consequently, they could be used as possible targets for immune checkpoint inhibitors or targeted therapy. However, despite successful efforts to identify the mutations involved in ACPs, the results obtained with targeted therapies in ACPs are not encouraging, and their application in current management remains very limited [29].

Conversely, molecular mutations found in PCPs have a more powerful implication in clinical practice and represent a new weapon in the management of these tumors. Brastrianos et al. found that a BRAF V600E mutation is detected in 94.6% of PCPs (36/39 tumors using genomic sequencing) [45]. This genetic finding, which drives the oncogenesis in about 7% of all human cancers, seems to be involved also in other brain tumors, such as pleomorphic xanthoastrocytoma (70%), ganglioglioma (50%), and epithelioid glioblastoma (50%) [23]. The BRAF V600E mutation constantly activates the B-RAF serine/threonine kinase and the cascade involving the RAS/RAF/MEK/ERK pathway. This well-known mechanism of “signal transmission” mediates the cellular responses to growth signals through the modulation of cell proliferation, differentiation, and cell survival [48,49,50,51,52]. In PCPs, the expression of oncogenic BRAF V600E is observed in most of the tumor cells but the activation of the MAPK pathway is rather restricted to a few tumor cells. The causative effect of BRAFV600E mutation is not clear, but it seems to involve a proliferative advantage to tumor SOX2+ cells with a mutated differentiation potential [1,2,13,15].

The detailed knowledge of this network represented the basis for the development of targeted therapy against different human solid (such as melanoma, colorectal carcinoma, and NSCLC) and blood (i.e., hairy cell leukemia) cancers [50,52]. With regard to brain tumors, both B-RAF (dabrafenib and vemurafenib) and MEK inhibitors (trametinib and cobimetinib) seem to present an optimal pharmacokinetic profile with good therapeutic concentration in the CNS and are currently used in the treatment of some glial tumors, such as pleomorphic xanthoastrocytoma, ganglioglioma, and BRAF V600E–mutant glioblastoma with systemic metastases [13,15]. Furthermore, molecular targeted therapy is opening new frontiers for the treatment of aggressive recurrent PCPs and, more recently, for a neo-adjuvant approach directed to selected cases of suspected PCPs with high operative risk [16]. A summary of the potential molecular targets and respective treatments for CPs is reported in Table 2.

### 4.3. Adjuvant Treatment

The main application and the well-reported results of targeted therapy involve recurrent PCPs. For these cases, BRAF and MEK inhibitors represent a rescue strategy against aggressive lesions resistant to standard treatments, as well as an adjuvant treatment to surgery and radiotherapy in the case of recurrence.

Aylwin et al. first described the effects of a single-agent anti-BRAF therapy (i.e., vemurafenib 960 mg BID for 7 months) against a multi-operated and multi-irradiated PCP [21]. Despite an initial excellent response (i.e., volume tumor reduction of 95%), the tumor recurred 6 weeks after the end of the treatment, showing the limited role of this therapy on long-term tumor control. Brastrianos et al. reported a case of large recurrent cystic PCP, which showed a volume reduction of 80% (81% cystic and 85% solid volume) after 35 days of combined treatment with trametinib and dabrafenib [22]. The drastic tumor volume reduction allowed for further endoscopic resection and an adjuvant radiosurgery, improving the clinical and radiological picture at the last follow-up (18 months). Interestingly, BRAF V600 mutation was also detected in the peripheral blood of the patient. However, since this patient underwent several operations, it is unclear if the surgery trauma caused DNA to be released into the peripheral blood, and to date this remains the only case where the mutation was found by a simple liquid biopsy. Recently, De Stefano et al. [25] and Khaddour et al. [23] reported a successful tumor control after a partial endoscopic resection followed by both BRAF/MEK inhibitor therapy and adjuvant irradiation (radiotherapy or radiosurgery) for two recurrent PCPs. Specifically, a combined therapy with dabrafenib (150 mg BID oral) and trametinib (2 mg QD oral) was administered after a endoscopic transsphenoidal surgery for 5 and 9 months, respectively, providing a dramatic tumor reduction (94% and 70%, respectively). Finally, the treatment was completed with SRS-Gk (25 Gy, iso 50%) in the first case and with radiotherapy (52.2 Gy/29 Frz) in the second case. These studies confirm that “the medical tumor debulking” obtained with BRAF and MEK inhibitors needs to be integrated with standard approaches (surgery and/or radiotherapy) conventionally performed in case of recurrent PCPs.

Several reports showed that a combined targeted therapy, with both BRAF and MEK inhibitor agents, seems to have a better efficacy in terms of tumor volume reduction and rapidity of actions than a single BRAF inhibitor [16,17,18,19,20,22,23,25,58,59]. Rostami et al. reported an excellent tumor control in a recurrent V600E BRAF mutated PCP after a treatment of 7 weeks with dabrafenib and trametinib [59]. The patient was initially treated with a single-agent therapy (dabrafenib) for 3 weeks, providing a tumor volume reduction of only 11%. Following the addition of trametenib, the tumor volume showed an impressive reduction (91%) at 8 weeks follow-up (thus, after 5 weeks of combined therapy) confirming a potential synergistic action of a combined therapy on the magnitude and rate of response. Furthermore, Brainstein et al. recently showed that combined treatment reduces the risk of cutaneous toxicity and resistance developing in a series of different BRAF-mutated brain tumors [20]. Furthermore, a phase III clinical trial in metastatic melanomas showed a better progression free survival and overall survival in patients who received a combined therapy with MEK and BRAF inhibitors than in those treated with single-agent therapy [60]. Even if the mechanism involved in this synergistic effect is not clear, it is possible to argue that the presence of an additional downstream inhibition along the MAP-K cascade could increase the total effect of the blockage and reduce the possibility of molecular escape strategies [61].

The efficacy of the combined target has been recently confirmed in an adjuvant setting of a phase-2 clinical trial (NCT03224767) where vemurafenib and cobimetinib were administrated to 16 patients with newly diagnosed PCPs, and treated in 28-day cycles [60]. The authors found an excellent response rate (i.e., RR: 14/16 patients, 93.75%, CI: 68–99.8%), with a mean volume tumor reduction of 83%. Regardless of the surgical technique used (which is not specified in this study), the results support the administration of the targeted therapy immediately after a subtotal surgical resection, with the aim of delivering radiation therapy on a smaller target volume.

Regardless of the type of treatment (i.e., single agent or combined therapy), the BRAF/MEK inhibitor therapy is associated with a response rate of almost 100% with a tumor volume reduction ranging from 80–90% after 3–4 months of treatment. To the best of our knowledge, the literature reports only one case of resistance to medical treatment (one of the 16 patients included in the ongoing phase-2 clinical trial (NCT03224767)). Even if a selection bias should be considered, the efficacy and rapidity of action justify the introduction of this therapy as the first line option in rescue or adjuvant treatment and as upfront therapy in selected cases.

Of course, all of these studies present several limitations. To date, only case reports or short case series have been published and many relevant points, such as long-term outcomes, delayed secondary effects from this treatment, as well as the more appropriate timing of the administration, require further confirmation in larger series.

### 4.4. Neo-Adjuvant Treatment

The very encouraging results obtained with the combined BRAF and MEK inhibitors in patients with recurrent and aggressive PCPs is opening new frontiers for a neo-adjuvant approach to this condition. Calvanese et al. first reported a case of a solid third ventricular mass presenting with headache, psychiatric disorders, and left optic atrophy [16]. The authors decided to perform a trans-ventricular neuro-endoscopic biopsy, revealing a PCP harboring the conventional BRAF V600E mutation. The patient underwent the combined targeted therapy with dobrafenib and trametinib, with a volume reduction of 90% at 4 months follow-up, followed by radiotherapy at the end of oncological treatment (Figure 2). To date, this is the only case of a PCP managed by a neo-adjuvant approach. Obviously, obtaining a diagnosis of PCP by non-invasive (or at least minimally invasive) techniques represents a mandatory step to apply a neo-adjuvant treatment. Fujito et al. considered that three preoperative clinical factors could be able to predict the BRAF mutation in PCPs with a high sensitivity and specificity: age above 18 years, a supradiaphragmatic tumor location and absence of intratumoral calcification [62]. However, the administration of such a specific treatment needs a formal diagnosis of BRAF V600E mutated PCP which cannot be fulfilled only with radiological findings. In our original article, we proposed a new algorithm to apply in case of radiological suspicion of PCP, but also in the presence of giant and invasive CPAs with difficult surgical access [16] (Figure 3). In these patients, a tissue sample obtained by different mini-invasive approaches (endoscopic transventricular, endoscopic endonasal or even by mini-craniotomy tailored on the lesion) should be the first option prior to making clinical decisions. This strategy should be considered, even in case of neurological deficit (such as visual impairment), due to the rapid and impressive results in reducing tumor volume offered by oncological treatment. It is important to highlight that medical therapy alone seems to be insufficient in providing a long-term control of the lesion. In other words, the aim of the neo-adjuvant approach should not be “curative” but rather “complementary” to radiotherapy or surgery. Indeed, several reports found that stopping medical treatment after a good initial response could be associated with a non-negligible risk of early tumor recurrence in patients with a previously diagnosed PCP [2,13,14,16,18,19,21,38,39]. Moreover, an excellent long-term tumor control seems to be reached when radiotherapy or radiosurgery are performed at the end of oncological treatment [5,63]. Of course, a lower tumor volume also strengthens the possibility to obtain a gross total resection by surgery [4,16,27,28,64]. Therefore, surgical resection has to be considered, as well as radiotherapy to integrate and complete the ideal management of the patient diagnosed with PCP and initially treated with targeted therapy.

## 5. Conclusions

Management of CPAs remains very challenging due to the long-term morbidity and the high recurrence rate. Surgery and radiotherapy remain the only effective treatments to obtain tumor control. However, new frontiers opened by targeted therapies (mainly in PCPs) are switching the management forward toward a new neuro-oncological perspective, based on molecular pathways and targeted therapies. In addition to the BRAF mutation, which is strongly associated with papillary forms, the role of inflammatory mediators and immune checkpoint pathways could also open new field of treatment possibilities by the administration of drugs currently used for other pathologies. Obviously, this will require a multidisciplinary approach, including an oncologist or a new type of physician both trained in oncology and endocrinology.

According to the review of the most recent literature, the BRAF/MEK inhibitor therapy should be considered in any case of PCP due to its efficacy and rapidity of action. Furthermore, our work highlights the importance of this treatment, not only in the adjuvant setting, but also as a first line of treatment (a neo-adjuvant approach). Indeed, in the case of recurrent PCP, the tumor shrinking provided by a combination of BRAF and MEK inhibitors as targeted therapy might reduce the morbidity associated to conventional therapeutic options. Most original, for selected cases of surgically challenging CPs, our proposal is first to perform a simple biopsy. In case of a confirmed PCP diagnosis, the authors support the use of anti BRAF/MEK therapy for a tumoral debulking followed by radiotherapy or surgery.

However, many questions remain. Firstly, in case of a neo-adjuvant setting, the application of the targeted therapy requires a biopsy, as liquid biopsy or radiology cannot ensure a reliable diagnosis. Secondly, in neurologically impaired patients (hydrocephalus or severe visual deficit), a neo-adjuvant therapy can be difficult to apply at this level of evidence. Thirdly, even if treatment seems to be well tolerated, the ongoing phase-2 clinical trial (NCT03224767) reported the need to stop the treatment for three patients due to adverse reactions. Finally, the limited number of cases reported opens several questions concerning the duration of the treatment, as well as the potentially different results on the solid and cystic parts of the tumor. Further studies with larger series are required to answer these points.

## Figures and Tables

**Figure 1 brainsci-13-00515-f001:**
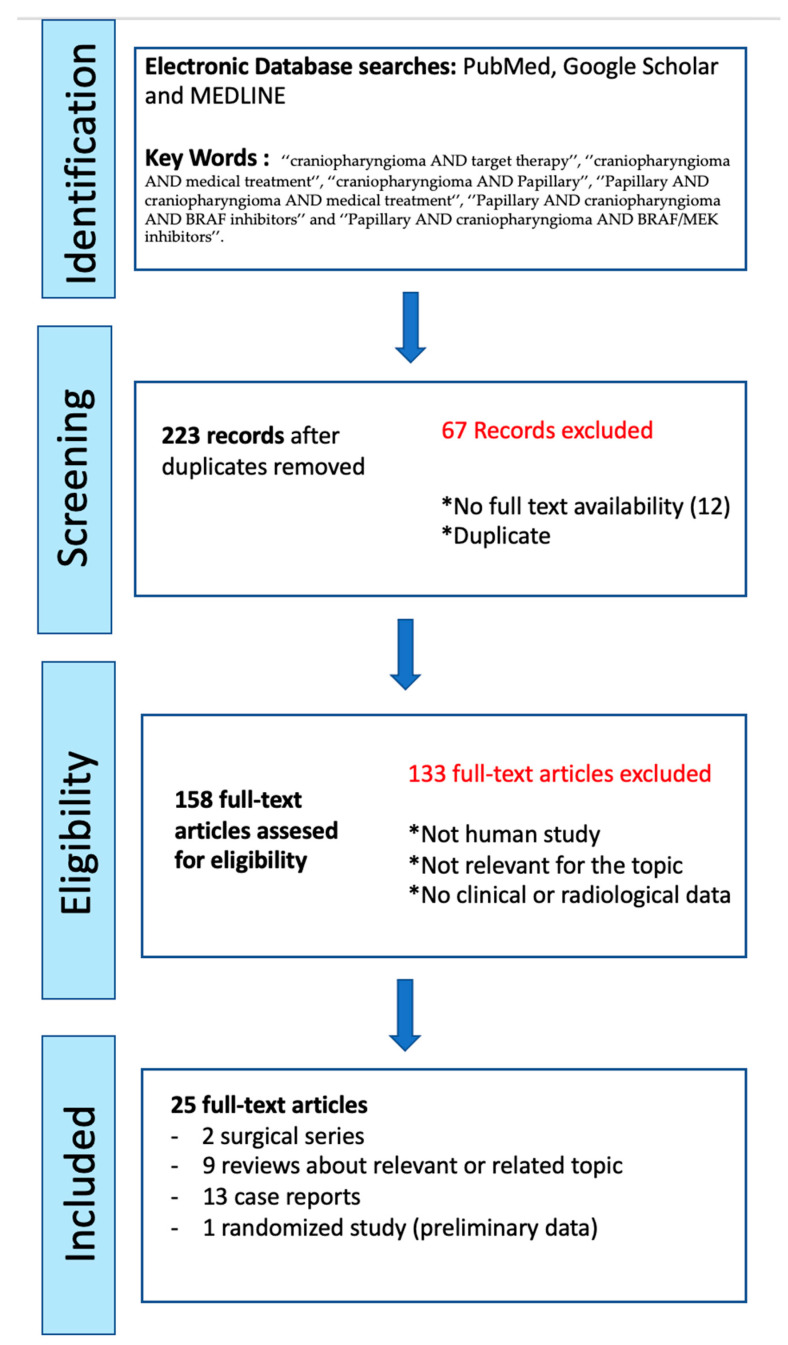
PRISMA flow chart of the literature review. * Exclusion criteria for full-text articles assessed for eligibility.

**Figure 2 brainsci-13-00515-f002:**
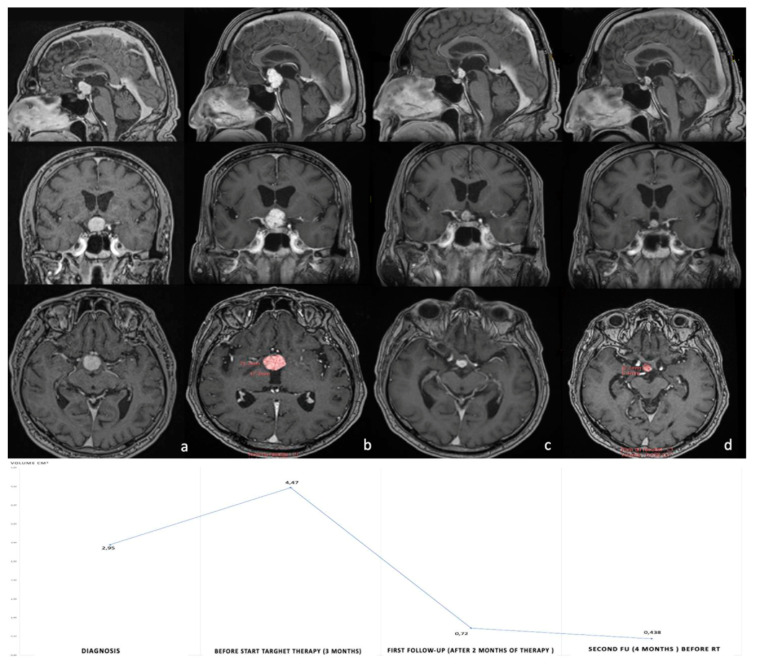
Post-gadolinium axial, coronal and sagittal T1WI MRI images of the first reported case treated with neoadjuvant BRAF/MEK inhibitor therapy. (**a**) Post-contrast T1-weighted image shows the large homogeneously enhanced intraventricular mass measuring 19 × 18.5 mm maximal axis and 2.945 cm^3^ volume. (**b**) Shows the progression of the intraventricular tumor portion after trans-ventricular endoscopic biopsy (18% of tumor volume). Panels c and d show a dramatic reduction in volume at 2 months (**c**) and 4 months (**d**) after commencing combined BRAF/MEK inhibitor treatment. Note the complete resolution of the mass effect on suprasellar neurovascular structures and on the Monro foramen. Volume curve has been reported in the inferior part of the figure. RT, radiotherapy. *Taken from with permission*: Calvanese, F., et al., Neoadjuvant B-RAF and MEK Inhibitor Targeted Therapy for Adult Papillary Craniopharyngiomas: A New Treatment Paradigm. *Front Endocrinol* (Lausanne), 2022. 13: p. 882381 [16].

**Figure 3 brainsci-13-00515-f003:**
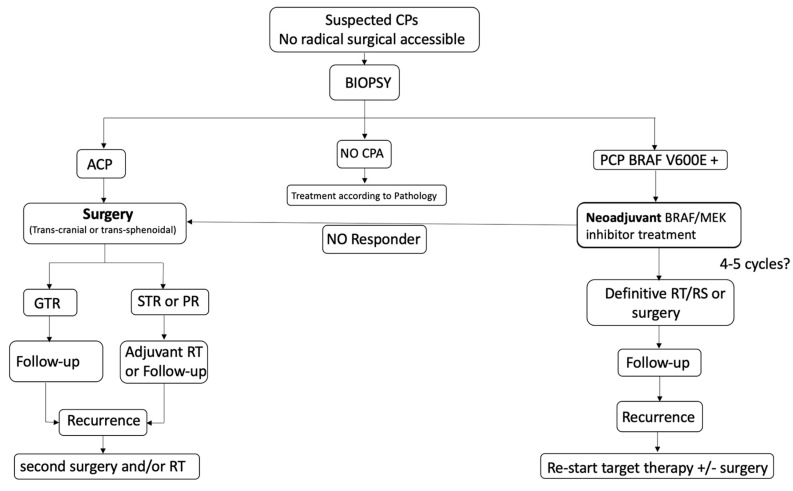
Proposed management algorithm in the case of ventricular and tubero-infundibular CPA which are not good candidates for a safe radical resection. CP, craniopharyngioma; ACP, adamantinomatous CPA; PCP, papillary CP; RT, radiotherapy; RS, radiosurgery. *Taken from with permission:* Calvanese, F., et al., Neoadjuvant B-RAF and MEK Inhibitor Targeted Therapy for Adult Papillary Craniopharyngiomas: A New Treatment Paradigm. *Front Endocrinol* (Lausanne), 2022. 13: p. 882381 [16].

**Table 1 brainsci-13-00515-t001:** Radiological predictors of the hypothalamic involvement in CPs.

Radiological Findings	Imaging Method	High-Risk Factor of Hypothalamic Involvement and Grading
Topographic location according to Pascual et al. classification [37,38]	Sagittal T1 and T2 WI	Tubero-infundibular and secondary intraventricular
Tumor morphology [37,38]	Sagittal and coronal T1WI-Gd	Elliptical, multilobulated tumors
PS morphology [37,38]	Sagittal T1 and T2 WI	Not visible or amputated/infiltrated
Relationship TVF/tumor [37,38]	Coronal T2WI	Middle third > upper third > bottom third
Hypothalamic oedema	Axial T2WI/FLAIR	Moustache appearance [36]
Hypothalamic oedema	Coronal T2WI/FLAIR	Grade B and C according to Hayashi et al. [32] T2WI hypothalamic changes according to Van Gompel et al.’s grading [21]
Irregular contrast enhancement according to Van Gompel et al.’s grading [34]	Coronal T1WI-Gd	Grade 1: Irregular contrast enhancement or hypothalamic changes on T2WI Grade 2: Irregular contrast enhancement associated with hypothalamic changes on T2WI
Hypothalamic involvement according to Saint-Rose and Puget’s grading [31]	Sagittal MRI	Grade 1: Hypothalamic compression Grade 2: Infiltration or unidentifiable hypothalamus
Hypothalamic and MB involvement according to Muller’s grading [39]	Sagittal MRI	Grade 1: No hypothalamic involvement Grade 2: Anterior hypothalamic involvement (i.e., no MB involvement) Grade 3: Anterior and posterior hypothalamic involvement including MB
CP adherence to hypothalamus according to Prieto et al.’s classification [8] MRI variables predicting the risk of adherence: 1. Location of Hypothalamus/TVF * 2. PS 3. Morphology	Coronal and sagittal T2WI	Level I or mild risk: Sellar/suprasellar (leptomeningeal layer) Upper third hypothalamic level, no visible PS, round or pear-like morphology Level II or moderate risk: Intraventricular (fibrovascular stem with ependyma) Lower third hypothalamic level, visible PS, round shape Level III or serious risk: Suprasellar/pseudo-ventricular (blown-like shape adherences) Upper third hypothalamic level; No visible PS; elliptical, dumbbell and multilobulated morphology Level IV or severe risk: Tubero-infundibular or secondary intraventricular (ring-like shape or wrapping paper-like adherences) Middle third hypothalamic level, No visible PS, elliptical morphology Level V or critical risk: Replacement of TVF Middle third hypothalamic level, no visible PS, elliptical and multilobulated morphology

* Location of the hypothalamus or TVF considering the tumor mass on the mid-sagittal and coronal T2WI.PS: pituitary stalk; TVF: third ventricular floor; MB: mamillary body.

**Table 2 brainsci-13-00515-t002:** Potential molecular targets and corresponding medical treatments for CPs.

Molecular or Genetic Alterations	CPs Type (% of Case)	Tumoral Compartment or Specific Tumoral Cell	Molecular or Biological Disarrangement	Possible Molecular Treatment
BRAF_V600E_	PCP (95–100%)	SOX2+/Height proliferating progenitor tumor cell	Hyperactivation MAPKK pathway	BRAF/MEKi [16]
CTNNB1 (Exon 3 Beta catenin gene)	ACP (70–100%)	Single or cell clusters (SOX2+) in tumor mass or at the base of epithelial protrusion	Hyperactivation WNT/beta catenin pathway leading to SASP	Senolytics [53] Wnt/β-catenin signaling inhibitors (several on-going trials) [54]
CTNNB1 (Exon 3 Beta catenin gene)	ACP (70–100%)	Forefront or leading edge of the tumor	Hyperactivation MAPK pathway (possible crosstalk with WNT pathway)	Trametinib or combined BRAF/MEKi [29]
CTNNB1 (Exon 3 Beta catenin gene)	ACP (70–100%)	Single or cell clusters (SOX2+) in tumor mass or at the base of epithelial protrusion (ACP recurrence)	SHH secretion and hyperactivation of SHH pathway IL-1 and IL-6 secretion Secretion of VEGF, FGF2, TGF beta and increased expression of PDGFR-alpha Hyperactivation of EGF/EGFR pathway (AREG, EGFR, and ERBB-3)	SHH pathway inhibitors (vismodegib) [1] IL-1R inhibitor (anakinra) Antiangiogenic drugs [55]: * bevacizumab * Selctive-PDGFR-alfa blockers (ripretinib) TKI: cetuximab, erlotinib, and lapatinib [15]
CTNNB1 (exon 3 Beta catenin gene)	ACP (70–100%)	Single or cell clusters (SOX2+) in tumor mass or at the base of epithelial protrusion	MMP9 and MMP12 overexpression [56] LCK, EPHA2, SRC overexpression [56]	MMP9/12 inhibitor AZD1236 dasatinib
PD1/PDL-1	ACP (100%) PCP (100%)	Cyst-lining in ACP, and to basal tumor cells in PCP	Immunomodulatory action	ICI [49]
PIK3CA and the TSC2 mutations [57]	ACP (-) PCP (-)	_	Hyperactivation of mTor pathway	Everolimus (mTor inhibitors) Copanlisib (pan-PIK3 inhibitor)

LCK: lymphocyte-specific protein tyrosine kinase; EPHA2: ephrin type-A2; SRC: proto-oncogene tyrosine-protein kinase Src; SHH: sonic hedgehog; TKI: tyrosine kinase inhibitors; ICI: immune checkpoint inhibitors; BRAF/MEKi: BRAF (ex. dabrafenib) and MEK (ex. trametinib) inhibitor agents; SASP: pro-tumorigenic senescence-associated secretory phenotype; mTor: mammalian target of rapamycin; PIK3CA: phosphatidylinositol 3-kinase; TSC2: tuberous sclerosis complex 2.

## Data Availability

Not applicable.

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
