# Peer review of "Current Advances in Papillary Craniopharyngioma: State-Of-The-Art Therapies and Overview of the Literature"

_brainsci, 2023, doi:10.3390/brainsci13030515_

Round 1

Reviewer 1 Report

I would like to thank the authors for their review, which comprehensively describes approaches to the treatment of papillary craniopharyngioma. I have the following questions: 

 1. Are there reports on the possibility of using liquid biopsy to detect the BRAF V600E mutation? According to the authors, can this approach be used to diagnostic papillary craniopharyngioma? What are the limitations of this approach? 

2. How often is targeted therapy used in the treatment of papillary craniopharyngioma? Is the administration of BRAF and/or MEK inhibitors justified in patients with papillary craniopharyngioma? What are the limitations of its use? Duration of therapy? Have cases of resistance been observed with vemurafenib, trametinib and dabrafenib?

Author Response

Reviewer #1 general considerations: I would like to thank the authors for their review, which comprehensively describes approaches to the treatment of papillary craniopharyngioma.

Authors response: We would like to thank the reviewer for the positive feedback on our work.

Please find our detailed responses to the reviewer comments for our manuscript in the file attached 

Reviewer 2 Report

This topic is very interesting, but paper needs to be improved. Look at these points:

- Authors wrote "We will particularly focus on radiological 64 and genetic spectrum of PCPs". It is better to highlight in the text where these results are present. Paragraph no. 3.

- How were the papers of this review selected? Please add a PRISMA flow diagram.

- Method section is missing. It is not clear how many papers were selected. Please revise and clarify.

- Lines 88-90: "Hayashi et al. reported that the degree of 88 extent of peritumoral hypothalamic edema on coronal FLAIR and T2WI MRI images 89 could be an index of pre- and postoperative functional outcome in adult patients."Consider also these 2 refs.: - Resolution of Papilledema Following Ventriculoperitoneal Shunt or Endoscopic Third Ventriculostomy for Obstructive Hydrocephalus: A Pilot Study. Medicina (Kaunas). 2022 Feb 13;58(2):281. doi: 10.3390/medicina58020281  ---  Radiological and endocrinological evaluations with grading. Pituitary. 2019 Apr;22(2):146-155. 

- Conclusion must be improved. As there are several similar papers and reviews to this one, what does this paper want add new to the literature? Improve conclusion section.

- According to the authors, what can be the evolution of the treatment of craniopharyngiomas in the next future?

Author Response

Reviewer #2 general considerations: This topic is very interesting, but paper needs to be improved.

Authors response: We would like to thank the reviewer for the positive feedback on our work and for his suggestions.

Please find our detailed responses to the reviewer comments for our manuscript in the file attached . 

Round 2

Reviewer 2 Report

Authors solved all my criticisms.